# Origin of fungal hybrids with pathogenic potential from warm seawater environments

Valentina del Olmo [1,2], Verónica Mixão [1,2,7], Rashmi Fotedar [3], Ester Saus [1,2], Amina Al Malki [3], Ewa Księżopolska [1,2], Juan Carlos Nunez-Rodriguez [1,2], Teun Boekhout [4] & Toni Gabaldón [1,2,5,6] ✉

Hybridisation is a common event in yeasts often leading to genomic variability and adaptation. The yeast *Candida orthopsilosis* is a human-associated opportunistic pathogen belonging to the *Candida parapsilosis* species complex. Most *C. orthopsilosis* clinical isolates are hybrids resulting from at least four independent crosses between two parental lineages, of which only one has been identified. The rare presence or total absence of parentals amongst clinical isolates is hypothesised to be a consequence of a reduced pathogenicity with respect to their hybrids. Here, we sequence and analyse the genomes of environmental *C. orthopsilosis* strains isolated from warm marine ecosystems. We find that a majority of environmental isolates are hybrids, phylogenetically closely related to hybrid clinical isolates. Furthermore, we identify the missing parental lineage, thus providing a more complete overview of the genomic evolution of this species. Additionally, we discover phenotypic differences between the two parental lineages, as well as between parents and hybrids, under conditions relevant for pathogenesis. Our results suggest a marine origin of *C. orthopsilosis* hybrids, with intrinsic pathogenic potential, and pave the way to identify pre-existing environmental adaptations that rendered hybrids more prone than parental lineages to colonise and infect the mammalian host.

Since the 1980s, the number of outbreaks of emerging infectious diseases - those rapidly spreading or appearing for the first time in an affected population - has increased about 6.7% each year[1]. Although often neglected, the emergence of new fungal pathogens has been increasingly recognised as a serious threat to human health and society[2–4]. The impact of humans on natural environments, globalisation, widespread use of fungicides in agriculture, as well as climate change are some of the factors influencing the emergence and dispersal of new fungal pathogens[5,6]. In addition, several studies report the isolation of pathogenic human-associated yeasts from non-clinical sources, suggesting that the ecology of these species is complex and that the environment could serve as a reservoir of opportunistic fungal pathogen species[7–9]. Despite their relevance, we still know very little about how new fungal pathogens emerge. Knowing what evolutionary routes new pathogens have followed, how relevant virulence traits were acquired, or what factors facilitated their colonisation of the human host or their opportunistic infections, could be instrumental for the monitoring and prevention of future deadly outbreaks.

[1]Life Sciences Department. Barcelona Supercomputing Center (BSC), Jordi Girona, 29, 08034 Barcelona, Spain. [2]Mechanisms of Disease Program, Institute for Research in Biomedicine (IRB), The Barcelona Institute of Science and Technology, Barcelona, Spain. [3]Department of Genetic Engineering, Biotechnology Centre, Ministry of Municipality and Environment, P.O Box 20022 Doha, Qatar. [4]College of Science, King Saud University, Riyadh, Saudi Arabia. [5]ICREA, Pg. Lluis Companys 23, Barcelona 08010, Spain. [6]Centro de Investigación Biomédica En Red de Enfermedades Infecciosas, Barcelona, Spain. [7]Present address: Bioinformatics Unit, Infectious Diseases Department, National Institute of Health Dr. Ricardo Jorge, Av. Padre Cruz, 1649-016 Lisbon, Portugal. ✉e-mail: toni.gabaldon@bsc.es

Recently, hybridisation was proposed as a possible evolutionary route towards the emergence of new fungal species with pathogenic potential[10]. A scenario supported by the finding of the hybrid nature of several *Candida* pathogens, including *Candida orthopsilosis*[11,12], *Candida metapsilosis*[13,14], *Candida tropicalis*[15], *Candida theae*[16] and, most remarkably, *Candida albicans*[17]. Although the possibility of an origin of new pathogens via hybridisation is intriguing, many questions remain open that require the identification of the hybrid parentals. Here we shed light on the origin of one such pathogenic hybrid: *C. orthopsilosis*.

The *Candida parapsilosis* species complex comprises four closely related species capable of causing infections in humans: *Candida metapsilosis, Candida orthopsilosis, Candida parapsilosis sensu stricto*[18] and *Candida theae*[16,19]. Importantly, *C. metapsilosis, C. orthopsilosis* and *C. theae* have been found to be of hybrid nature[11–14,16]. The vast majority of sequenced clinical strains from *C. metapsilosis* (100%) and *C. orthopsilosis* (87.5%) are hybrids isolated from multiple locations in different continents indicating a worldwide distribution. Hence, parental lineages are never or very rarely found in these settings, which prompted researchers to hypothesise that the pathogenic hybrids might have arisen from non- (or less) pathogenic parentals that may thrive in unknown environmental niches[12,13,16,20]. In this view hybridisation might have enhanced the emergence of hybrid lineages with an advantage to thrive in new environments, such as in the human host. Most (48 out of 49) *C. metapsilosis* strains studied to date descend from the hybridisation between unknown parentals, referred to as A and B[11,14,21], whereas an additional, recently described strain descends from a different hybridisation event involving parental A and a third (also unknown) parental lineage (named C)[14]. The two analysed *C. theae* hybrid strains (one clinical and one from an industrial drink) descend from independent hybridisation events between the same two parental lineages, which also remain undescribed[16,19]. In the case of *C. orthopsilosis*, isolates sequenced so far originated from one of four independent hybridisation events between the same two parental lineages[12]. In 2014, Pryszcz and colleagues[11] described strain Co90-125 as one of the parents of the *C. orthopsilosis* hybrids (referred here as parental A) whereas isolates belonging to the parental B lineage have not been found to date.

Our view on the origin of hybrids with pathogenic potential is severely limited by the scarcity of isolates of environmental origin. Clinical isolates provide a partial view, representing only those lineages that successfully colonised and infected humans. However, several questions remain unanswered such as whether the hybrids originated in human-related environments or elsewhere, or whether phenotypic traits relevant for human colonisation result from environmental adaptations. To fill these gaps for *Candida orthopsilosis*, we sequenced and analysed the genomes of *C. orthopsilosis* strains isolated from the environment. Our results show that the majority of isolates are hybrids, which expands the map of ecological distributions where *C. orthopsilosis* hybrids can be found to include aquatic environments. There, hybrids appear to outnumber parentals, suggesting an advantage not only in the clinical setting but in some environmental niches too. We hypothesise that the genomic features that make hybrids highly competitive in certain environments might be also advantageous in other niches like the human body. Consistent with this statement, our phylogenetic reconstructions show that the marine environmental hybrids fall in (or close to) previously defined clades that harbour clinical isolates. Our study unexpectedly led to the discovery of the long-sought parental B lineage of *C. orthopsilosis*, thus providing a complete overview of the two parental lineages that gave rise to all known hybrids of this species. Moreover, we report phenotypic differences between the two parental lineages, such as thermotolerance, which have been inherited by hybrids and might play a role in pathogenesis. Finally, we observe intrinsic pathogenicity potential in parental and hybrid environmental isolates which disagrees with previous hypotheses of avirulent parental lineages and suggests that the

prevalence of hybrids in the clinics might be related to other factors not necessarily related to the ability of infection.

## Results

### Prevalence of hybrid strains in the marine environment

To broaden our knowledge on the diversity, distribution, and origin of *C. orthopsilosis* hybrids, we sequenced and analysed the genomes of nine *C. orthopsilosis* marine isolates. These were retrieved from four different coastal locations of Qatar harbouring warm waters with temperatures ranging from 35 to 44 degrees C (Supplementary Data 1). To assess the potential hybrid nature of the isolates, we determined the frequency and distribution of k-mers in the sequencing reads, as described earlier for other hybrids[14,17,22], see Methods). The k-mer frequency in non-hybrid strains is expected to follow a distribution with a single peak, while hybrid strains generally present two peaks, one corresponding to heterozygous regions and another one, with twice as much coverage, representing blocks of homozygosity. Our results (Fig. 1a), indicate that seven of the nine marine isolates displayed the double-peak pattern, suggesting a hybrid nature, while two showed a single peak indicating that they were non-hybrid strains.

We then mapped the Illumina short reads of all nine *C. orthopsilosis* marine isolates to the genome of the known parent A (Co90-125) and performed variant calling analysis (see Methods). We found that most marine isolates show an amount of homozygous variants that is in line with what has been reported in previous studies for clinical isolates (a total of 166,000 – 214,000 homozygous variants and a mean of 15 homozygous SNPs per kb). However, the two non-hybrid (SY36 and SY78) strains had more than double homozygous SNPs (492,000 homozygous variants, 40 homozygous SNPs per kb). Moreover, we determined the amount of heterozygous variants per kb which revealed that the two non-hybrid strains are also highly homozygous with 0.55 – 0.57 heterozygous SNP per kb (Supplementary Data 2), which is expected for a non-hybrid and in stark contrast with most hybrid strains studied up to date, which show heterozygosity levels of 7 – 20 heterozygous SNP/kb[12,23]. These results indicate that *C. orthopsilosis* isolates SY36 and SY78 are highly homozygous, non-hybrid strains.

After hybridisation, the divergence between the two parental haplotypes in heterozygous regions in the hybrid is expected to be similar across the genome. We analysed the density of the sequence divergence at heterozygous positions of the nine marine isolates. The seven putative hybrid strains displayed a single peak, at the same amount of divergence for all strains, indicating that in these strains, the two haplotypes descend from the same hybridisation event. The remaining isolates (SY36, SY78), did not show a single peak in the density plot which further supports a non-hybridisation scenario (Fig. 1b).

These results for environmental isolates are reminiscent of previous findings for *C. orthopsilosis* clinical isolates showing that hybrids are more prevalent than their homozygous parentals, although the ratio of non-hybrid strains may be higher in this sampling (2/9, as compared to 5/40 in previously sequenced *C. orthopsilosis* isolates). Thus, our results expand the geography and ecosystems where *C. orthopsilosis* hybrids can be found to include warm marine environments, and suggest that, also there, hybrids may be more adapted than their parentals.

### Identification of *Candida orthopsilosis* parent B lineage

All *C. orthopsilosis* hybrids studied to date descend from at least four independent hybridisation events between the same two parental lineages[12]. As mentioned above, two of the marine isolates from Qatar, named SY36 and SY78, showed k-mer profiles typical of non-hybrid strains (Fig. 1a). Importantly, their k-mer distribution plots also showed that a significant fraction of k-mers was absent from the reference genome. These results, together with the high number of homozygous

SNPs when compared to the parental A reference genome (see above) and their co-isolation with hybrid strains, strongly suggest that these homozygous marine isolates represent the missing parental B of *C. orthopsilosis* hybrids.

After a hybridisation event, alleles from each of the parentals can be visualised as heterozygous regions in the resulting hybrid. Thus, if strains SY36 and SY78 were in fact parent B, one would expect that, in heterozygous variants of hybrid strains, one of the alleles would match that of parent A and the alternative allele would match the base in parent B for that specific position. We tested this hypothesis using the *C. orthopsilosis* marine isolates in this study and available whole genome sequencing datasets from three previous studies[11,12,23], see Methods). The final dataset comprised 49 strains, 42 of which are hybrids, 5 strains correspond to parental A lineage and 2 strains potentially corresponding to parental B lineage (SY36, SY78). Given the slightly higher number of homozygous variants in strain SY36 (511,113)

compared to SY78 (508,494) and lower number of heterozygous ones (7214 in SY36 versus 8471 in SY78), we chose the former as a putative reference of parent B lineage, as it was the more homozygous and divergent from parent A. We evaluated the percentage of shared alleles with parentals A or B in heterozygous positions for all hybrid strains. The results showed that for an average of 89.77% of the heterozygous SNPs in a hybrid strain, when one of the alleles matched the reference allele in parent A, the alternative allele matched the homozygous SNP of SY36. Moreover, we also observed that, when encountering a homozygous SNP in the hybrids, in 92.06% of the cases, on average, the allele matched the base in parent B (Fig. S1; Supplementary Data 3). In sum, our analysis strongly suggests that SY36 is a highly homozygous non-hybrid strain that represents the missing parental B lineage of *C. orthopsilosis*.

A previous study by Schröder et al.[12] established that the mitochondrial genome of the *C. orthopsilosis* strains studied so far could be

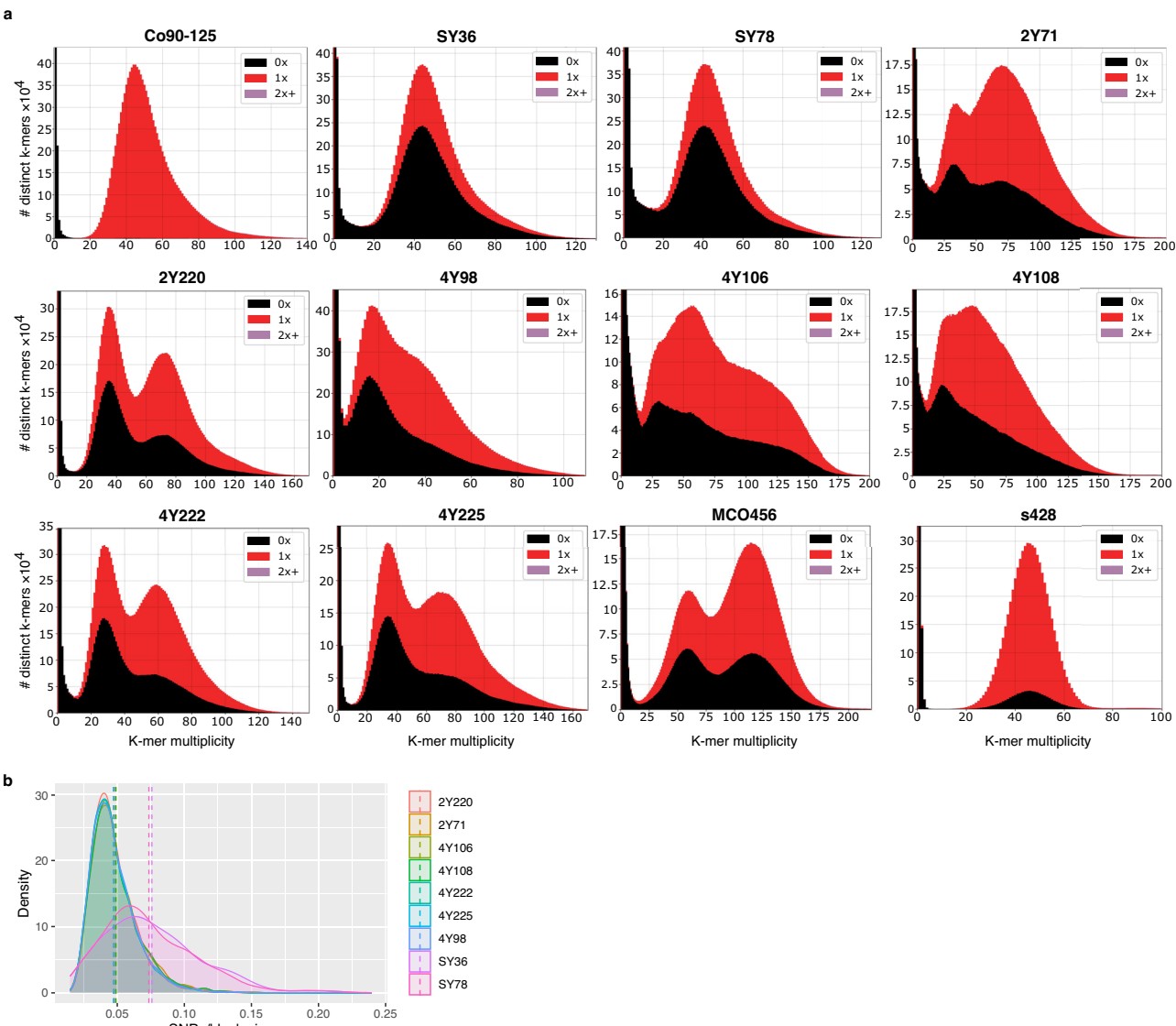

**Fig. 1 | Hybrids and the parental B lineage amongst *C. orthopsilosis* marine isolates. a** Frequency of 27-mers of *C. orthopsilosis* known parental A strain Co90-125, marine non-hybrid SY36 and SY78 strains as well as hybrid marine strains 2Y721, 2Y220, 4Y98, 4Y106, 4Y108, 4Y222 and 4Y225. Previously known clinical hybrid strain MCO456 and clinical non-hybrid strain s428 (A-lineage) have been added to this figure for comparison purposes. The presence of a single peak indicates non-hybrid strains. Isolates showing a two-peaked profile are hybrids. The

first peak (with half of the coverage) corresponds to heterozygous positions, whereas the second one represents homozygous regions. Colours in the plot represent the presence (red) or absence (black) of the k-mers in the reference genome of parent A Co90-125 (ASM31587v1[64]). **b** Plot of the density of sequence divergence in heterozygous blocks (larger than 100 base pairs) of marine *C. orthopsilosis* strains. Source data are provided as a Source Data file.

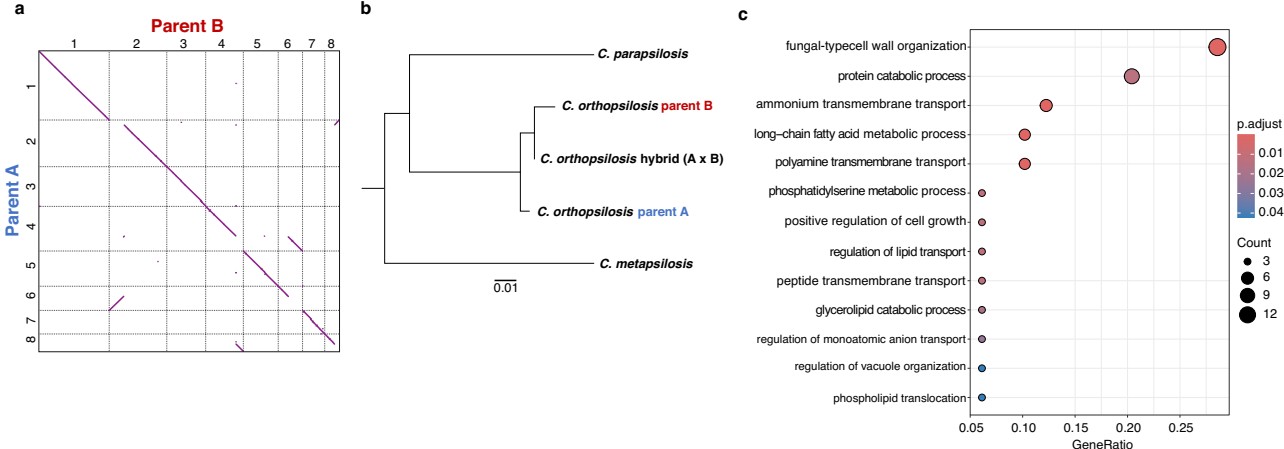

**Fig. 2 | Reference genome assembly of novel *C. orthopsilosis* parent B. a** Dot plot assessing the similarity between assemblies of Co90-125 parent A ASM31587v1[64] and newly identified and assembled parent B strain SY36 (see Methods: Genome assembly *C. orthopsilosis* parent B). **b** Species tree showing the phylogenetic relationships between members of the *C. parapsilosis* species complex. *C. orthopsilosis* strain MCO456 was chosen as a representative of hybrids while Co90-125 and SY36 represent parents A and B, respectively. **c** GO term enrichment of Biological Process terms of genes harboured in genomic regions that are parent B specific. GO term enrichment analysis was done using clusterProfiler v. 3.14.3, which performs the hypergeometric test. Source data are provided as a Source Data file.

**Table 1 | Calculation of divergence between *C. orthopsilosis* parental lineages A and B**

| Reference genome | Sequencing reads (alternative parental) | Homozygous variants | # bases in reference genome covered by > 30 reads of alt. parent | % divergence between reference and alternative parental |
| --- | --- | --- | --- | --- |
| Co90-125 (parent A) | SY36 (parent B) | 510,485 | 12,368,794 | 4.127 |
| SY36 (parent B) | Co90-125 (parent A) | 514,680 | 12,590,099 | 4.088 |

classified in four different mitotypes (mt1 to mt4) being mt4 the mitotype corresponding to parent A (Co90-125). According to this study, nuclear clades 1 and 3 likely received their mitotypes (mt1 and mt3) from their parent species A. The study also identified two recombinant mitotypes named mtR1 and mtR2, which are the result of two separate events of inter-lineage recombination between mt4 and mt2. Finally, the research suggests that strains harbouring mitotype 2 likely received it from parent species B. We generated a phylogeny based on variants from the mitochondrial genome including all 49 considered isolates. Our results recapitulate and confirm the four main mitotypes (mt1-4) and the two recombinant ones mtR1 and mtR2 (Fig. S2). Notably, the mtDNA of all environmental hybrids clustered in already defined mitotypes (mt1, 2 and 3). As expected, isolates SY36 and SY37 corresponding to parental B lineage were found placed close to mt2. This result suggests that recombination events between the two parental mitotypes (mt4 for parent A and mt2 for parent B) likely gave rise to the recombinant mitotypes mtR1 and mtR2 and further supports the notion that SY36 and SY78 represent the parental B lineage.

### A reference genome assembly for *Candida orthopsilosis* parent B and comparison with parent A

Our analysis so far has unveiled the existence of the parental B lineage, unknown until now. We then used a combination of long oxford nanopore (ONT) and short Illumina reads sequencing technology to generate a genome assembly of isolate SY36, as a reference for the parental B lineage (see Methods). The resulting assembly comprised eight scaffolds, one per chromosome, and had a total size of 13,172,296 bp. We found that the genomes of both parental strains (Co90-125 and SY36) are largely collinear (Fig. 2a). The genome divergence between the two parental lineages was, on average, 4.1% (Table 1, see Methods), a value slightly lower than previous estimations (4.5–5%) based on the density of the divergence of SNPs in

heterozygous blocks[11,23]. We then generated a species tree depicting the phylogenetic relationships between members of the *Candida parapsilosis* species complex: *C. parapsilosis sensu stricto*, *C. metapsilosis*, *C. orthopsilosis* parental lineages A and B as well as a representative of a hybrid strain (MCO456). In agreement with the results presented so far in this study, the branch giving rise to the *C. orthopsilosis* members further branches into three clearly different groups: the two parental lineages at both ends and the hybrids, placed in between both parental branches (Fig. 2b). Given the observed divergence between parentals we sought to find genomic regions of difference, exclusive from each parental lineage. We mapped the Illumina reads from one parent to the assembly of the other (and vice versa) and searched for non-coverage regions. Regions larger than 1000 base pairs with no coverage were identified along both genomes as parental-specific blocks (regions of N repetitions were excluded from this analysis). Interestingly, we found that B-specific regions (present in both parental B strains SY36 and SY78) were significantly enriched in fungal-type cell wall organisation, storage vacuole and plasma membrane components (Figs. 2c and S3). No GO term enrichment was found amongst proteins encoded in A-specific regions.

### Phylogenetic network reconstruction shows close relationships between *C. orthopsilosis* hybrid and parental lineages

To avoid reference-related biases and have both parental haplotypes equally represented in the phylogenetic tree, we generated a phylogeny based on variants (Supplementary Data 4) where the sequence of each of the strains included a concatenation of two alignments, one taking parental A and another taking B as reference. The resulting network tree showed the two parental lineages as the most distant branches and all hybrid strains distributed in intermediate positions clustered into four distinct groups (Fig. 3a), corresponding to previously defined hybrid clades[12,23]. Our approach can recapitulate the phylogeny, which is now complete and includes the two parental

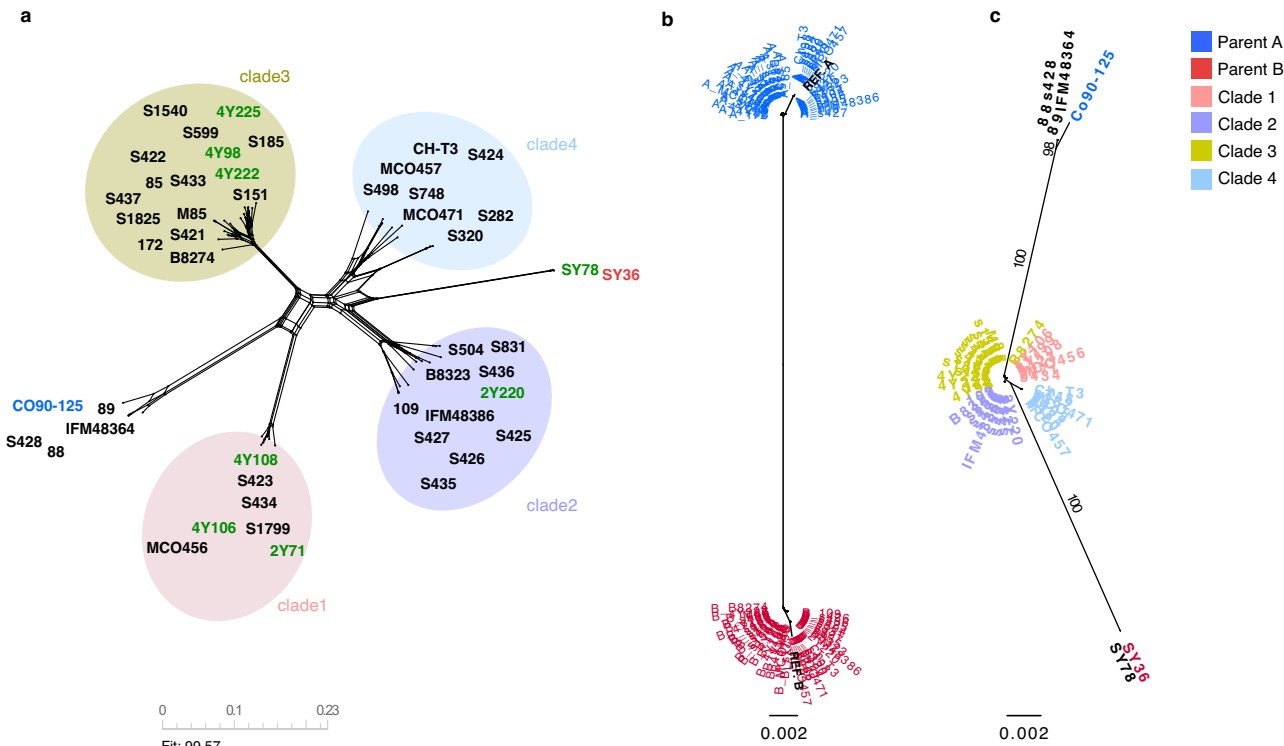

**Fig. 3 | Phylogenetic relationships between *C. orthopsilosis* parental and hybrid lineages. a** Neighbour net splits tree network based on alignments harbouring homozygous variants of all *C. orthopsilosis* strains. Different hybrid clades are circled in colours. Parent A Co90-125 is highlighted in blue, parent B SY36 in red and all other marine isolates in green. **b** Maximum-likelihood tree showing phased haplotypes A and B for each hybrid strain at heterozygous positions. A haplotypes are highlighted in blue and B haplotypes in red. **c** Phylogenetic tree based on heterozygous sites. Sequence alignments of phased haplotype A and B were concatenated for each strain in order to build the tree. Different hybrid clades as well as parental reference strains are highlighted in colours. Source data are provided as a Source Data file.

lineages that gave rise to all hybrid strains. Regarding the position of the parental lineages, on the one hand the branch giving rise to the parent A lineage emerges from between clades 1 and 3, while parent B lineage emerges between clades 2 and 4. Interestingly, far from clustering together or forming a different clade, phylogenetic reconstruction placed the environmental marine isolates within already defined clades (2Y71, 4Y106 and 4Y108 isolates in clade 1, 2Y220 in clade 2 and 4Y222, 4Y225 and 4Y98 in 3), closely related to the rest of clinical isolates suggesting a shared origin and little genome rearrangement after the initial hybridisation (Fig. 3a).

*C. orthopsilosis* hybrid isolates display very different patterns and distribution of SNPs with some clades being significantly more heterozygous than others (Fig. S4). In addition, we reconstructed both haplotypes (A and B) at heterozygous loci shared by all hybrid strains. A maximum likelihood tree shows a clear separation between the two haplotypes (Fig. 3b) further confirming SY36 as parental B and corroborating that all hybrids originate from the same two parental lineages. Finally, we concatenated the sequences of haplotypes A and B for each strain and built a maximum likelihood tree based on the sequence of shared heterozygous blocks (Fig. 3c). The resulting tree recapitulates the tree built based on homozygous variants with the parental lineages being the two most distant branches, parent A emerging between clades 1 and 3 and parent B between clades 2 and 4.

*C. orthopsilosis* can harbour two mating-type-like (*MTL*) idiomorphs, *MTL*a and *MTL*α, that can occur in two types each (Type 1 and Type 2) that differ in approximately 5% in sequence, each corresponding to one of the two parental lineages (Sai et al., 2011, Schröder et al., 2016). So far, all isolates belonging to parent A lineage have been found to be homozygous for *MTL*a Type 1[12,23]. We found that both the marine isolates belonging to the parent B lineage were also

homozygous for *MTL*a but showed the alternative type (Type 1) (Fig. S5). Parent B contributed with *MTL*a in hybrids of clade 3 (homozygous *MTL*a/a) and 4 (heterozygous *MTL*a/α). Marine hybrids belonging to clade 1 (2Y71, 4Y106 and 4Y108) were *MTL*α/α and exhibited the same genotype as the rest of clinical isolates from the same clade. Marine hybrids from clade 3 (4Y98, 4Y222 and 4Y225) were homozygous *MTL*a inherited from parent B, as for most of the clinical isolates belonging to this clade. Clade 2 is the most heterogeneous in terms of *MTL* genotypes with strains being homozygous *MTL*a/a or *MTL*α/α and heterozygous *MTL*a/α, yet marine isolate 2Y220 differed from all the rest of the clinical isolates from its clade. *MTL* idiomorphs a and α also carry alleles of a poly(A) polymerase gene (*PAP*), an oxysterol binding protein gene (*OBP*), and a phosphatidylinositol kinase gene (*PIK*) which can also have idiomorphs a or α[24]. All *C. orthopsilosis* strains sequenced to date have the same genotype in the *MTL*, *PAP*, *OBP* and *PIK* genes (a/a, α/α or a/α). However, 2Y220 was found to be *MTL*a/α but *PAP*a/a *OBP*a/a. The *PIK* gene had undergone LOH resulting in a chimeric gene being homozygous a/a at the 5′ and heterozygous a/α at the 3′ of the gene (Fig. S5). These results indicate that, after hybridisation, the MTL locus has been extensively and recurrently shaped by LOH in patterns that vary from strain-specific to clade-specific, which hint at different timings of these events.

### Loss of heterozygosity in *C. orthopsilosis* hybrids does not favour any of the two parental lineages

A common phenomenon after hybridisation is the appearance of genomic incompatibilities or redundancies between the two parental genomes frequently leading to detrimental consequences for cell fitness and survival. In order to stabilise their genome, hybrid cells undergo LOH by which a portion of one of the parental haplotypes is

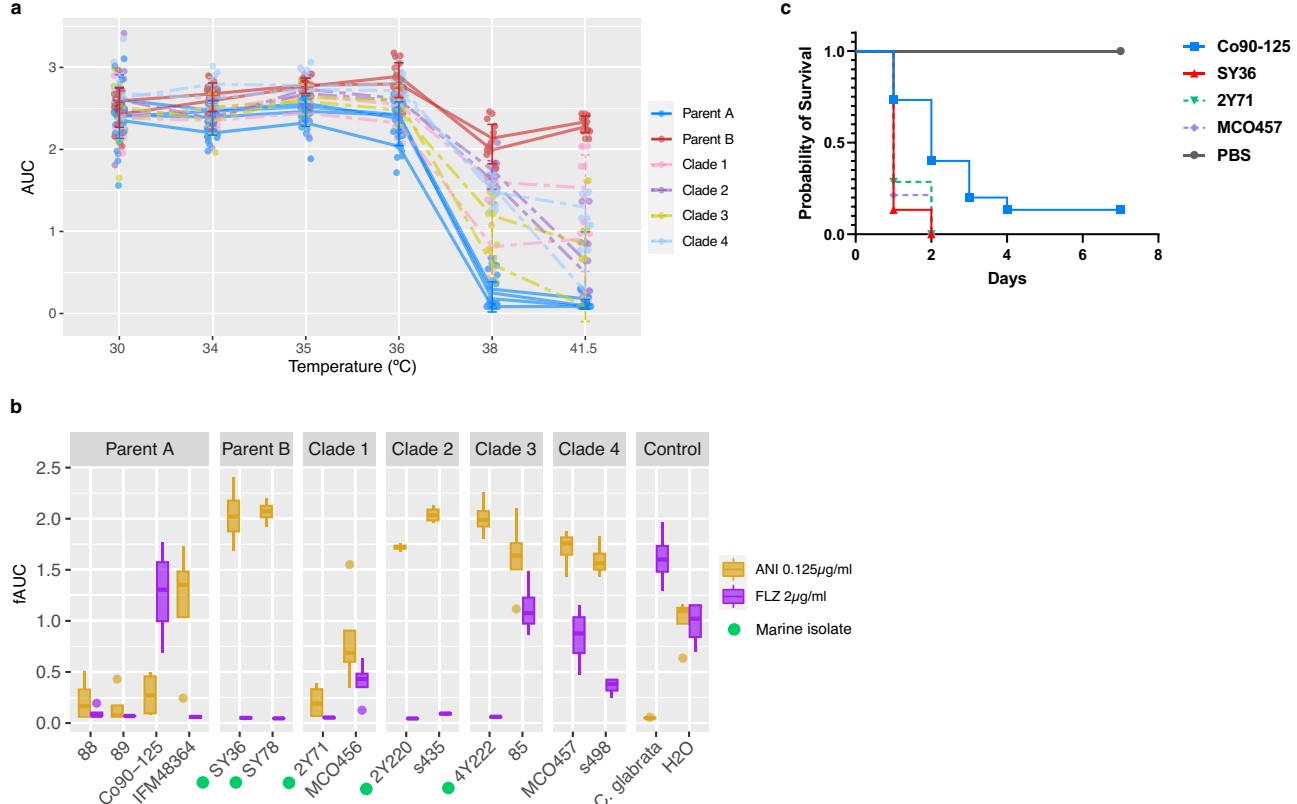

**Fig. 4 | Phenotypic differences between a selected subset of *C. orthopsilosis* strains under different conditions.** Area under the curve (AUC) inferred from measurements taken every 15 min during a period of 24 h in solid medium. **a** Area under the growth curve of *C. orthopsilosis* hybrid and parental strains (*n* = 8 biological replicates over 2 independent experiments) under temperatures ranging from 30 °C to 41.5 °C. Strains from the same clade or parental lineage are grouped together. Clades and parental lineages are represented in different colours. Data are presented as mean values +/− SD. **b** Fitness area under the curve (fAUC) of *C. orthopsilosis* hybrid and parental strains (*n* = 4 biological replicates) in the presence of anidulafungin (ANI) and fluconazole (FLZ). Green circles in the X-axis indicate marine isolates. The centre line in the boxplot indicates the median value, the boxes contain the Q1 and Q3 quartiles (IQR). The whiskers extend up to 1.5 × IQR. **c** Survival curves of *G. mellonella* injected with 10⁶ *C. orthopsilosis* cells per larva (*n* = 15) during a period of 7 days. Difference between strains was significant with a *P* < 0.001 by the log rank (Mantel-Cox) test. Source data are provided as a Source Data file.

lost and the region becomes homozygous for the other haplotype via recombination between homoeologous chromosomal regions or double-strand breaks followed by repair mechanisms[25–28]. We sought to study LOH events in *C. orthopsilosis* hybrids and assess the contribution of each of the parental lineages to such events. We inferred LOH in each of the strains, taking either parent A or parent B as reference, and plotted the distribution of LOH blocks and heterozygosity regions along the chromosomes (Fig. S6). In agreement with previous studies[12,23], we observed, for the most part, a clade-dependent distribution of LOH with patterns and amount of LOH varying substantially between each clade, confirming previous observations of four hybridisation events[12]. Contribution to LOH from each parent was equivalent in most strains, thus showing no clear tendency of a predominant haplotype in terms of LOH acquisition (Fig. S6; Supplementary Data 5). We then searched for GO term enrichment of genes harboured in LOH blocks inherited by each parent, shared by all hybrids, by hybrids from the same clade and by strains coming from either a clinical or an environmental source. No enrichment was found except for LOH blocks inherited from parent B shared by all strains belonging to clade 4 where cell wall and plasma membrane cellular components were overrepresented (Supplementary Table 1). Interestingly, 74% of genes harboured in these LOH regions were parent B-specific genes (which show no coverage by reads of parent A Co90-125 strain). In general, and in agreement with previous studies[12,23], no clear signs of functional selection were observed to happen upon LOH events.

## *C. orthopsilosis* parental B presents intrinsic thermotolerance, which is retained in all hybrids

Given the broad differences at the sequence level, the hybrid nature and the environmental or clinical origin, we sought to determine whether these were reflected at the phenotypic level. The environmental strains reported in this study were isolated from warm waters (35 to 44 °C)[29] but, the optimal temperature for the growth of most yeasts is approximately 30 °C[30–33]. Thus, we tested the ability of a set of representative parental and hybrid strains to grow at temperatures ranging from 30 to 41.5 °C in both solid and liquid medium. Our results show similar growth rates in the range of 30 to 36 °C. Strikingly, 35 °C was the optimum temperature for most strains (Fig. S7). As temperatures rose above 36 °C, the growth of all strains decreased, albeit to different degrees. Strains belonging to the parental A lineage had a significant reduction of growth and were completely unable to grow at temperatures above 40 °C. On the other hand, parental B strains were able to tolerate the heat and grow at 41.5 °C. Hybrid isolates showed an intermediate phenotype (Fig. 4a; Fig. S7). In order to explain this phenotype we assessed the genotype of heat-shock- and temperature-related proteins in the parental and hybrid strains but found no clear genetic difference (in terms of small or copy number variants) that could explain the lack of thermotolerance of parental A strains.

Considering that the environmental isolates were obtained from marine waters we assessed the tolerance of our representative strains to salinity. The mean PSU (practical salinity units) of the ocean is 35 PSU which corresponds to 0.6 M. However, PSU values around the

coastal area of Qatar, where the *C. orthopsilosis* marine strains were isolated, vary between 40 and 70[29,34]. Therefore, we tested the growth of our selected strains on medium containing concentrations of NaCl ranging from 0.5 to 5 M. Although the growth rate was negatively correlated to the salt concentration, we did not observe any differences in growth between the strains and no isolate was able to grow at salt concentrations of 2.5 M NaCl or higher (Fig. S7). In addition, we assessed the growth of the strains under reductive stress (DTT) and in the presence of calcofluor white (CFW) and SDS which cause cell wall and plasma membrane stress, respectively. No major differences were observed in the growth rate of the strains in the presence of DTT and CFW with the exception of M85 not showing any growth with CFW. Regarding membrane stressor agent SDS, it caused stains from clades 1 and 3 to be unable to grow. The growth of parent A Co90-125 was also impaired. The growth of parental B strains was the least affected by the presence of all stressing agents (Fig. S7).

## Parental and hybrid lineages exhibit distinct susceptibility to antifungal drugs

Given the clinical relevance of drug susceptibility and the important question of whether it is present among environmental strains, we assessed the susceptibility of a representative subset of strains to two antifungal drugs frequently used in clinical settings, anidulafungin and fluconazole. To account for phenotypic differences caused by the nature of the media, these drug susceptibility experiments were performed in both liquid and solid media. In the case of anidulafungin, all *C. orthopsilosis* isolates displayed a higher growth rate than the *C. glabrata* control (Figs. 4b and S8). This phenotype could be explained by a naturally-occurring proline to alanine mutation in *FKS1* HS1 (P653A) that has been previously observed in members of the *C. parapsilosis* species complex, which renders isolates less susceptible to echinocandins[35,36]. All strains tested displayed the above-mentioned mutation. Moreover, parental B strains showed a higher fitness and tolerance to the drug and strains belonging to the A lineage displayed the highest sensitivity. The phenotype of hybrid isolates was an intermediate state between the A and B although hybrids from clades 2, 3 and 4 showed higher fAUC (fitness Area Under the Curve; see Methods) than hybrids belonging to clade 1 in solid medium (Figs. 4b and S8). In liquid medium, hybrid strains from clade 1 (which in solid medium had shown poor degree of growth) as well as strain 85 from clade 3 were not able to proliferate.

Conversely, when grown in the presence of fluconazole, parent A strain Co90-125 was the least susceptible strain and the growth of parent B lineage was severely impaired. All environmental strains were completely unable to grow in the presence of fluconazole whereas most of the clinical ones tested (Co90-125, MCO456, 85, MCO457 and s498) could do so (Figs. 4b and S8). Growth trends were almost identical in both liquid and solid medium (Figs. 4b and S8). Of note, strains from the highly heterozygous clade 4 showed better growth rates in the presence of fluconazole suggesting that heterozygosity might be advantageous in these settings. Copy-number variant analysis revealed the existence of a duplication at the locus harbouring the MDR1 gene, exclusively in the fluconazole-tolerant parent A (Supplementary Data 6). We then examined the sequences of genes that have been previously related to fluconazole resistance and compared the genotypes between tolerant and susceptible strains (Fig. S9). We found that (aside from the MDR1 duplication in parent A) the vast majority of mutations were clade specific, with no mutations being exclusive of susceptible strains, hinting that the mechanisms that regulate tolerance towards fluconazole may vary among clades and might not be restricted to one single gene.

## Novel environmental parent B isolate shows pathogenic potential

Hybridisation has been shown to result in pathogenic hybrid offspring in other fungi such as *Aspergillus*[37], *Cryptococcus*[38] or *Malassezia*[39]

species. In the case of *Aspergillus*, the resulting hybrid showed increased pathogenic potential when compared to its parents[37]. Furthermore, the lower prevalence of parental lineages over hybrid strains in clinical settings has been previously hypothesised to be the result of reduced pathogenicity of the parental strains[10]. In order to test this hypothesis, we used a *Galleria mellonella* host infection model and monitored the survival of larvae infected with either a parental strain (parent A Co90-125 or parent B SY36) or a hybrid (2Y71 or MCO457). Additionally, we sought to test the effect of the environmental source by evaluating two strains isolated from clinical settings (Co90-125 and MCO457) and two others from the marine source (SY36 and 2Y71). Given that hybrids from clade 1 and clade 4 are the most different in terms of homo- and heterozygosity, we tested one hybrid from clade 1 (2Y71) and one belonging to clade 4 (MCO457). We observed significant differences between the survival curves of *G. mellonella* larvae infected with different strains (Fig. 4c) and found a tendency for parent A (Co90-125) to be the least virulent strain (although with significant *P*-value in only one replicate; Fig. S10). Previous studies have failed to identify correlations between levels of loss of heterozygosity in hybrids and pathogenicity[10,12]. Similarly, we did not observe any differences in virulence between the two tested hybrid strains harbouring different levels of LOH. Unexpectedly, the parent B strain SY36 was found to be as virulent as the two hybrids (2Y71 and MCO457).

Secretion of proteolytic enzymes has been previously linked to virulence in other pathogenic yeast species like *C. albicans*, *C. glabrata* or *C. parapsilosis*[40]. Aspartyl proteases secreted by these pathogens influence cell damage, phagocytosis and other processes associated with infection[41]. We assessed the ability of hybrid and parental isolates to secrete such enzymes and found that only parent A strain Co90-125 and, to a lesser degree, strains IFM48386 (clinical) and 2Y220 (environmental) from clade 2 were able to do so (Fig. S11).

Thus, these results point to an intrinsic pathogenic potential of environmental *C. orthopsilosis* strains as well as parental lineages hinting that the ability to colonise humans might be the secondary result of adaptations to thrive in another niche.

## Discussion

In this study we have examined the genomes of nine *C. orthopsilosis* marine isolates. Environmental strains from hybrid species in the *Candida parapsilosis* complex are highly underrepresented in the current available genomic dataset, which includes only one marine isolate *C. metapsilosis*[12,23] and another from *C. theae*[16]. Our research describes the genomes of environmental *C. orthopsilosis* isolates found at several subtropical marine sites.

Similarly to what has been reported in clinical settings[12,23], our data show a higher abundance of *C. orthopsilosis* hybrids over parental strains suggesting that hybrids are prevalent and might possess a fitness advantage over parental lineages not only in clinical settings but also in the marine ecosystem, although the dominance of hybrids is more acute in the clinical setting. However, our environmental dataset has a small size and thus, information about more environmental samples should be gathered in order to make such a statement with confidence.

One could expect that strains isolated from one specific niche would be genetically more similar (or have niche-specific features) when compared with a population of samples isolated from a completely different environment. In contrast, our analysis showed the new marine hybrid isolates clustering in distinct clades (1, 2 and 3) together with clinical isolates (Fig. 3). This finding indicates that clinical and environmental hybrids are closely related and suggests multiple independent human colonisations from the environment without the need of specific genetic adaptations to thrive in the mammalian host. It also shows that the sampled sites present a high genetic hybrid diversity, with different clades coexisting in the same location. Finally, these observations also suggest that genetic traits that make hybrids

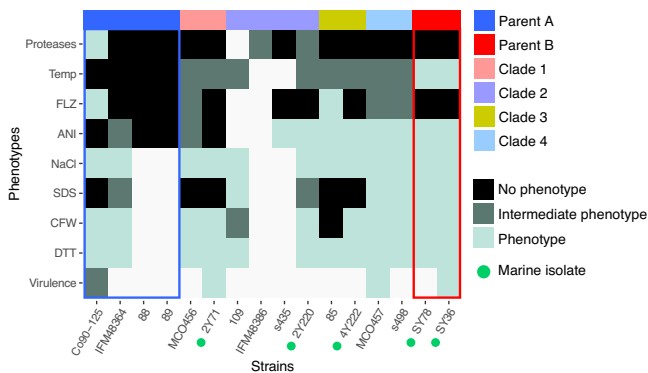

**Fig. 5 | *C. orthopsilosis* hybrids inherit different traits from parental lineages.** Overview of phenotypic differences between hybrid and parental lineages. The term "Phenotype" indicates growth of the strain under that specific condition in comparison to "No phenotype" which refers to absence of growth in that condition. The term "Intermediate phenotype" refers to a lower degree of growth than that of "Phenotype". Clades and parental lineages are indicated in different colours. Green circles indicate marine isolates. Source data are provided as a Source Data file.

competitive and able to thrive in the marine ecosystem might be advantageous to do so in a human-associated niche.

We found two strains (SY36 and SY78) among the nine isolates that were highly homozygous and genetically distant from the known parental A lineage. Our additional analyses provided a strong support for the affiliation of these strains to the long sought parental lineage B from *C. orthopsilosis*. This unexpected observation constitutes the most relevant finding of our survey. The presence of parental B in co-existence with several of the previously identified hybrid clades, suggests that the warm sea water environment could be a melting pot where these two lineages hybridise. Importantly, yeast species that pose a threat to human health have also been found in the marine environment as studies describe the isolation of opportunistic human pathogens *C. albicans* from oceanic waters[42] and *C. parapsilosis*, *C. tropicalis* and *C. glabrata* living in association with oysters and mussels collected from estuarine areas[29,43,44]. Moreover, the multidrug-resistant pathogen *Candida auris* has also been recently isolated from a marine ecosystem in a study that also reports finding a drug-susceptible isolate and raises a hypothesis about an environmental origin of this species[9]. However, broader sampling efforts may clarify whether this parental lineage and the resulting hybrids may thrive in alternative environments. We did not find any parental A isolate. This could be due to our small sampling size and a presumably lower abundance of parental A in this location. Alternatively, parental A and B may overlap in other locations or niches.

After hybridisation, loss of heterozygosity (LOH) events are an important source of genomic diversity[45-47]. In this study, we analysed the patterns and distribution of LOH in *C. orthopsilosis*. We could assign the origin of each LOH block to either parent A or B and found no predisposition of LOH blocks in hybrids to be inherited from either parental lineage. Nevertheless, and consistent with the notion of at least four independent hybridisation events[12], we found that the length of LOH blocks and their distribution across the genome of hybrid strains was clade-dependent, suggesting low levels of genome rearrangement after the initial hybridisation (Fig. S6).

The availability of both parental lineages A and B genomes allowed us to search for parental-specific loci in *C. orthopsilosis*. These loci in parent B are enriched in membrane and cell-wall related proteins which could potentially play a role in virulence, adhesion and pathogenesis (Figs. 2c and S3). In relation to this observation, parental lineage B strains appeared to have a higher growth rate than parent A in the presence of all stress-inducing agents tested (Fig. S7), anidula-fungin (Figs. 4b and S8) and at high temperatures (Figs. 4a and S7). We

also observed that parent B (SY36 strain) showed a degree of virulence similar to that of hybrid strains (Figs. 4c and S10), which suggests that hybridisation in *C. orthopsilosis* might not result in an increased pathogenicity like in other fungi but influence other aspects that render hybrids more prevalent than parentals in at least two niches (clinics and marine environment). Indeed, we observed that hybrids were not only genetically distinct to the parental lineages but also displayed phenotypic differences compared to them under several conditions (Fig. 5). In general, hybrid isolates displayed traits that resembled either parental lineage A or B but interestingly, some isolates showed intermediate phenotypes (between A and B) under conditions like temperature increase or presence of fluconazole which might influence adaptation to new niches.

Little is known about the ecology of pathogenic yeasts outside clinical settings[7]. However, the existence of marine fungi has been known since the late 1800s[48] and researchers published the first reports about their diversity in the early 60 s[49,50]. In the particular case of yeasts, studies report that species within *Candida* and *Rhodotorula* appear to be the predominant genera encountered from marine isolates[51,52]. From an ecological perspective, marine yeasts are able to degrade biomass and hydrocarbons as well as parasitize members of the macrofauna[48]. The optimum temperature for the growth and proliferation of most yeasts is approximately 30 degrees Celsius (°C)[30-33]. Only a reduced number of species are able to successfully proliferate at temperatures around or higher than 37 °C, making the mammalian basal temperature a potent first barrier of defence against fungal pathogens[53]. All environmental marine samples described in this study were isolated from subtropical waters at temperatures ranging from 35 to 44 °C (Supplementary Data 1). Our phenotypic analysis showed that the optimum temperature at which all *C. orthopsilosis* strains grew was 35 °C (Figs. 4a and S7), a value worryingly close to the mammalian thermal barrier that might explain the ability of these opportunistic pathogens to colonise the human host even during fever episodes. The intrinsic thermotolerance of parental B, a trait that parental A does not display, seems to have been inherited in part by all examined hybrid strains, which display an intermediate phenotype (Figs. 4a and 5). Similarly, studies report other filamentous fungi with the potential to develop thermotolerance overtime[54]. *C. auris* and its closest known relative *C. haemulonii* have been isolated from marine environments too[9,55]. Climate change and increasing global temperatures have been hypothesised to be a main driver in the emergence of this thermotolerant human pathogen. In relation to this hypothesis, our findings point to a possible environmental (marine) source of *C. orthopsilosis* which could have served as bed for hybridisation events. In agreement with the idea of an environmental origin, *C. orthopsilosis* strains have very recently been isolated from tea flowers in Thailand[56]. The environmental strains described in this study were not genetically distinct from clinical isolates suggesting that (*i*) *C. orthopsilosis* hybrid clinical strains likely represent multiple, independent colonisations from environmental sources and (*ii*) that adaptation to the marine ecosystem, and particularly to temperate waters, might render these strains capable of thriving at higher temperatures, which combined with a pre-existing pathogenic potential, allow colonisation and infection of humans. It is likely that these species existed as environmental fungi before being known as human pathogens. Global warming and increase in marine water temperatures might have acted as a selection force for thermotolerant *C. orthopsilosis* strains leading to their increased proliferation in this ecological niche. Expanding human presence worldwide could have facilitated niche expansion of these species or their vectors to new geographical locations. In consequence, thermotolerant strains might have been able to survive at the basal temperature of a mammalian host and expand especially in clinical environments where there is an enrichment of potential hosts with a depressed immune system, a second crucial barrier of defence against fungal infections. Moreover, the

results presented in this study show that, unlike some clinical isolates, marine strains present no tolerance to azoles (Fig. 4b). This observation suggests that environmental isolates might initially be sensitive to fluconazole. However, strains could potentially acquire resistance by recurrent exposure to the drug, acting as selecting pressure for the emergence of resistant strains. Therefore, aside from its already noticeable repercussions in the world's ecosystems, climate change poses a serious threat on human health by creating environmental pressures that select for fungi resilient to higher temperatures with the potential of becoming new human pathogens[3,6,57,58]. With increasingly rising global temperatures and overuse of antifungal drugs in agriculture and clinics[59], the appearance of environmental fungi able to breach the mammalian thermal barrier, acquire antifungal drug resistance and become potential pathogens is likely to increase.

## Methods

### Genomic DNA extraction, library preparation and sequencing

Genomic DNA extraction of *Candida orthopsilosis* strains was performed using the MasterPure Yeast DNA Purification Kit (Epicentre) following manufacturer's instructions with some modifications. Briefly, *Candida* cultures were grown from a single colony in an orbital shaker overnight (200 rpm, 30 °C) in 15 ml of YPD (Yeast Extract–Peptone–Dextrose) medium. Cells were harvested using 4 ml of each culture by centrifugation at maximum speed for 2 min, and then they were lysed at 65 °C for 15 min with 300 μl of yeast cell lysis solution (containing 1 μl of RNAse A). After being on ice for 5 min, 150 μl of MPC protein precipitation reagent were added into the samples, they were centrifuged at 16.000 g for 10 min to pellet the cellular debris and the supernatant was transferred to a new tube. In the case of samples used for Illumina sequencing, DNA was directly precipitated adding cold 100% ethanol, leaving the sample for 2 h at −20 °C and centrifuging them at 16.000 g for 30 min at 4 °C. The pellet was washed in 70% ethanol and left to dry. TE buffer was used to resuspend the DNA. Genomic DNA Clean & Concentrator kit (ZymoResearch) was used for the final purification. For the sample used to obtain long reads with Oxford Nanopore Technologies (ONT), *C. orthopsilosis* SY36 strain, 10 μl of RNase A/T1 mix (ThermoFischer Scientific) were added to the recovered supernatant and the sample was incubated at 37 °C for 2 h to ensure the complete removal of RNA. Then a phenol-chloroform purification step followed by a chloroform purification was performed using PLG (Phase Lock Gel) Heavy tubes: after 30 sec of centrifugation of the PLG tubes to pellet the gel, first the sample and then an equal volume of phenol-chloroform were added into the tube, vortexed and centrifuged at 16.000 g for 5 min. Then an equal volume of chloroform was added into the same tube, vortexed and centrifuged again at 16.000 g for 5 min. The final aqueous phase was recovered into a new tube, where DNA was precipitated using 100% cold ethanol and centrifuging the samples at 16.000 g for 30 min at 4 °C. The pellet was washed twice with 70% cold ethanol and, once the pellet was dried, the sample was resuspended in 50 μl of TE. All DNA samples were quality controlled for purity, quantity and quality, using NanoDrop Spectrophotometer (Thermo Fisher Scientific), Qubit dsDNA BR assay kit and a 1% agarose gel, respectively.

Libraries for Illumina whole-genome sequencing were prepared at the Functional Genomics Core Facility (FGCF) at the Institute for Research in Biomedicine (IRB Barcelona). *C. orthopsilosis* samples were sequenced in pools, as described in[60], with two other genomes of divergent species or genus (two of either *C. glabrata*, *C. albicans*, *Pochonia* or *Saccharomyces*). Crossmapper[61] was run prior to pooling in order to ensure the absence of cross-mapping reads between samples. For all samples, 500–1,000 ng of genomic DNA dissolved in a final volume of 50 μl TE buffer were sheared with a Bioruptor sonicator (Diagenode) using the following settings: temperature 4–10 °C; intensity: high; cycles: 3; cycle time: 5 min; cycle programme: 30 s pulse and 30 s rest time. At the end of each sonication cycle samples

were centrifuged at 4 °C and the water tank was refilled with precooled water. DNA fragmentation was quality controlled using a DNA High Sensitivity chip with the Bioanalyzer 2100 (Agilent) and quantified with the dsDNA HS assay using the Qubit fluorometer (Thermo Fisher Scientific). NGS libraries were prepared from 250 ng of fragmented DNA using the NEBNext Ultra II DNA library prep kit for Illumina, following the manufacturer instructions. All libraries were amplified through five PCR cycles using the NEBNext multiplex oligos for Illumina (New England Biolabs). The final libraries were quantified on Qubit (Thermo Fisher Scientific) and quality controlled in the Bioanalyzer 2100 (Agilent). An equimolar pool of libraries was prepared, and a final quality control by qPCR was performed. Libraries were sequenced on a NovaSeq6000 (Illumina) using the paired-end 150 nt strategy at the CNAG-CRG (Centre Nacional d'Anàlisi Genòmica - Centre de Regulació Genòmica). De-pooling and selection of sequencing reads was performed as described in[60] using the genome assemblies of *C. orthopsilosis* parent A and parent B concatenated as reference to separate the reads uniquely mapped to *C. orthopsilosis*.

Whole genome sequencing of long reads of *C. orthopsilosis* SY36 strain was performed at the CNAG-CRG (Centre Nacional d'Anàlisi Genòmica - Centre de Regulació Genòmica) using Oxford Nanopore Technologies (ONT). Genomic DNA was quality controlled on pulse field electrophoresis gel (Pippin Pulse, Sage Science) for DNA integrity, and the 260/280 and 260/230 ratios were used to detect the contamination on DNA samples with NanoDrop Spectrophotometer (Thermo Fisher Scientific). The sequencing library was prepared using the Ligation sequencing kit SQK-LSK109 from ONT. Briefly, 1.0 μg of the DNA was DNA-repaired and DNA-end-repaired using NEBNext FFPE DNA Repair Mix (NEB) and the NEBNext UltraII End Repair/dA-Tailing Module (NEB and followed by the sequencing adaptors ligation, purified by 0.4X AMPure XP Beads and eluted in Elution Buffer (SQK-LSK109). Two sequencing runs were performed on GridION Mk1 (ONT) using a Flowcell R9.4.1 FLO-MIN106D (ONT) and the sequencing data was collected for 110 h. The quality parameters of the sequencing runs were monitored by the MinKNOW platform version 3.6.5 in real time and base called with Guppy version 3.2.10.

### Raw sequencing data analysis

Raw sequencing reads and their quality were assessed with FastQC v0.11.9 (http://www.bioinformatics.babraham.ac.uk/projects/fastqc/). Trimmomatic v0.39[62] was used to filter out reads with quality below 10 or 4 bp sliding-windows with average quality per base of 20 and for the presence of adapters.

K-mer frequency plots were obtained using K-mer analysis Toolkit - KAT v2.4.2[63]. The default k-mer size of 27 was used for all the analyses. Additionally, KAT was also employed to assess the presence of each k-mer in the corresponding genome assemblies: *C. orthopsilosis* parent A Co90-125 ASM31587v1 [https://www.ncbi.nlm.nih.gov/datasets/genome/GCF_000315875.1/][64] or *C. orthopsilosis* parent B (see Methods: genome assembly *C. orthopsilosis* parent B).

### Read mapping, small variant calling and detection of CNVs

Sequencing reads were mapped to the following references: *C. orthopsilosis* Co90-125 parent A ASM31587v1[64], *C. orthopsilosis* parent B (see Methods: genome assembly *C. orthopsilosis* parent B) or *C. orthopsilosis* mitochondrial genome assembly NC_006972.1[65]. Given the high proportion of N-repetitions (1400.29 #N's per 100 Kb) present in the parent A Co90-125 ASM31587v1[64] genome assembly, we used Pilon v1.22[66] to correct it reducing the number of N repetitions from 1,400.29 to 320.87 #N's per 100 Kb. Augustus v3.2.3[67] was then used to annotate the new corrected parent A genome (using *C. albicans* as model) and compare it with the uncorrected version which showed a slight increase in the number of annotated proteins (from 5,383 in the original assembly to 5,443 proteins predicted in the corrected assembly). Completeness of both assemblies was evaluated with

BUSCO v4.1.4[68]. The corrected assembly showed 94.6% completeness compared to 94.5% from the uncorrected one.

Read mapping was carried out with BWA-MEM v0.7.17[69]. Picard integrated in GATK v4.1.9.0[70] was used to sort the resulting file by coordinate, mark duplicates, create an index file and obtain mapping statistics for all strains. The coverage of each sample mapped to the reference genome was determined with SAMtools v1.9[71]. Mapped reads were visually examined with IGV v2.8.13[72]. SAMtools v1.9[71] and Picard integrated in GATK v4.1.9.0[70] were also used to index and create a dictionary for each of the reference genomes. Levels of ploidy were estimated using nQuire histotest[73]. All hybrid strains were clearly diploid (nQuire histotest $r^2$ values raging 0.87 – 1.00). Ploidy estimation was inconclusive for strains belonging to parental lineages due to their low levels of heterozygosity (nQuire histotest $r^2$ values raging 0.00 – 0.24).

PerSVade v.1.02.4[74] module *call_small_variants* was used to perform variant calling. Minimum chromosome length was set to 5000 bp, minimum coverage to 20 and ploidy was set to 2 (diploid). PerSVade variant calling module performs small variant calling using three different programmes (GATK v4.1.2.0[70], Freebayes v1.3.1[75] and Bcftools v1.9[76]). Only variants passing quality filters in at least two out of the three above-mentioned programmes were considered for further analysis. Small variants were annotated using the module *annotate_small_vars*. Copy-number variants were detected and annotated using PerSVade modules *call_CNVs* (using a window size of 500 bp), *integrate_SV_CNV_calls* and *annotate_SVs* with default parameters.

## Genome assembly of *C. orthopsilosis* parent B

In order to generate a genome assembly for strain SY36 as a representative of *Candida orthopsilosis* parent B lineage, we employed an in-house pipeline which combined assemblers that use short Illumina and long ONT sequencing data. As mentioned in the previous section, Illumina reads were trimmed using Trimmomatic v0.39[62] and then assembled with Platanus v1.2.4[77]. Nanopore reads were filtered for quality with NanoFilt v2.7.1[78] and adaptors were removed using Porechop v0.2.3[79]. A subset of long reads with coverage 25X (rather than the complete set of reads) was used to build the assembly. Canu v1.8[80] was used to correct the ONT reads which were then assembled with DBG2OLC v20180222[81] using the Platanus and WTDBG2 v2.1[82] assemblies as reference. Ragout v2.2[83] was used for scaffolding. Correction of each individual as well as the final genome assembly was carried out with Pilon v1.22[66]. Augustus v3.2.3[67] was used to annotate the genome taking *Candida albicans* as a model species and assembly completeness was evaluated with BUSCO v4.1.4[68]. The quality of the assembly was assessed with Quast v5.0.2[84] as well as with the K-mer analysis Toolkit - KAT v2.4.2[63]. The presence of telomeric repeat sequences (5'-GGTTAGGATGTAGACAATACTGC-3') at the terminal regions of each chromosome was also used as control and confirmation of chromosome boundaries. Finally, Last v2.31.1 (https://github.com/lpryszcz/last) was used to assess similarity between the corrected parent A Co90-125 ASM31587v1[64] and parent B SY36 assemblies.

## Estimation of divergence between *C. orthopsilosis* parental lineages

The parental lineage A of *C. orthopsilosis* is represented by Co90-125 ASM31587v1[64] whereas lineage B is represented by SY36 (see Methods: genome assembly *C. orthopsilosis* parent B). The two parental assemblies as well as the short Illumina reads of each of the parent representative strains were used to calculate genome divergence between the two parental lineages. Short reads from Co90-125 (parent A) were mapped to the SY36 assembly (parent B) and vice-versa. Variant calling was carried out as described previously (Methods: Read mapping, small variant calling and detection of CNVs). For each parent, genome divergence was calculated dividing the number of homozygous variants by the size of the mappable genome covered by more than 30

reads (Table 1). We performed GO term enrichment analysis for genes harboured in parent A and parent B specific regions using cluster-Profiler v3.14.3[85], which performs hypergeometric test, for "Biological process", "Cell compartment" and "Molecular function" categories at Level 6.

## Phylogenetic analyses

The phylogenetic reconstruction based on homozygous variants of *C. orthopsilosis* mitochondrial genome was constructed as follows. Illumina reads from all isolates were mapped to *C. orthopsilosis* mitochondrial genome assembly NC_006972.1[65] followed by variant calling (see Methods: Read mapping, small variant calling and detection of CNVs) and substitution of homozygous variants in the respective reference genome. Bedtools subtract v2.30.0[86] was used to remove positions with heterozygous variants or INDELs in at least one strain from the final alignment. Then, for each strain an in-house script was used to substitute the homozygous variants in the reference genome. Positions covered by less than 20 reads were excluded from the analysis. The resulting concatenated alignment was used to build a maximum-likelihood tree using IQTree v2.0.3[87] with the automated best-fitting parameter. All trees were visualised with FigTree v1.4.4 (http://tree.bio.ed.ac.uk/software/figtree/).

For the nuclear genome, the phylogenetic reconstruction based on genome-wide polymorphisms was built using an in-house script that allowed homozygous variants to be substituted in the respective reference genome as well as to exclude genomic regions covered by less than 20 reads which had been previously detected using bedtools genomecov v2.30.0[86]. In order to have both haplotypes A and B represented in the phylogeny, the sequences resulting from alignments of each sample mapped to both parent A and B were concatenated. We previously established that the alleles present in heterozygous variants in hybrid strains originate from parent A and B. Thus, in the alignments for each strain, by assuming heterozygous variants equal to that of the reference genome and only substituting the homozygous variants in the reference genome we could obtain a non-biased representation of alleles A and B once the alignments of that strain mapped to both parents were concatenated. The length of the final concatenated alignment was 1.090.768 nucleotides. Splitstree v5.2.25[88] was used to compute the phylogenetic network and visualise the relationships between all isolates.

In order to generate a phylogenetic tree based only on heterozygous variants that would be equally representative for both parental haplotypes, we used JLOH v0.15.1[89], see Methods: Detection of loss of heterozygosity and heterozygous blocks) to identify heterozygous blocks (larger than 500 base pairs) shared by all 42 *C. orthopsilosis* hybrid strains using parent B as the reference strain. We next used HaploTypo v1.0.1[90] to infer the two haplotypes in the shared heterozygous blocks and generate two separate files containing haplotypes A and B separately for each of the samples. Positions tagged as unsolved were discarded from the phylogeny. Sequences of haplotype A and B were then concatenated. MAFFT v7.475[91] was used to align the sequences followed by trimming with TrimAl v1.4.rev15[92] using the best-automated method parameter. We obtained a final alignment of 11.368 nucleotides. IQTree v2.0.3[87] was used to generate a maximum-likelihood tree using the automated best-fitting parameter with bootstrap set to 1000.

The species tree including the members of the *Candida parapsilosis* species complex (*C. parapsilosis* strain CDC317, *C. metapsilosis* strain BP57, *C. orthopsilosis* strain Co90-125 as representative of parent A lineage, *C. orthopsilosis* strain SY36 as representative of parent B lineage and *C. orthopsilosis* MCO456 as representative of hybrid strains) was built using the sequence of four selected marker genes as mentioned in[93]. The sequence of ortholog genes for the different species represented in the tree was retrieved from the PhylomeDB v5

database[94]. Sequence alignment, trimming and building of a maximum-likelihood tree were performed as mentioned in the paragraph above.

### Detection of loss of heterozygosity and heterozygous blocks
Definition of LOH and heterozygous blocks was performed using JLOH v0.15.1[89] hybrid mode with parameters "--snp-distance" set to 200 and "--min-length" to 750. Regions of low coverage and low genome divergence (represented as LOH labelled as "REF" in the parental strain not used as reference) were labelled as ambiguous and excluded from the analysis in all samples. LOH and heterozygous blocks were plotted across the *C. orthopsilosis* genome using an in-house R script.

Enrichment analysis was performed using FatiGO[95] to find over represented GO terms in genes harbouring LOH or in chimeric genes.

### Phenotypic assays
Phenotypic tests were carried out with a selected subset of *C. orthopsilosis* strains. We used strains from each parental genome (Co90-125, IFM48364, 88 and 89 for parent A, and SY36 and SY78 for parent B) and two strains representative of each hybrid clade (2Y71 and MCO456 for clade 1, 2Y220 and 109 for clade 2, 4Y222 and M85 for clade 3, MCO457 and s498 for clade 4). For the hybrid samples of clades 1,2 and 3, we used one isolate retrieved from clinical settings and another from the marine environment. In the case of clade 4, only clinical samples were selected due to the lack of environmental strains belonging to this clade. *C. glabrata* strain CBS138 was used as control.

The experiments were performed as described in Nunez-Rodriguez Juan Carlos, Miquel-Àngel Schikora-Tamarit, Ewa Ksiezopolska, and Toni Gabaldón. "Q-PHAST: simple, large-scale quantitative phenotyping and antimicrobial susceptibility testing." (in preparation). Briefly, strains were streaked in individual YPD plates and grown overnight at 30 °C to obtain single colonies. Four single colonies, representing four biological replicates of each strain were picked, inoculated in 500 μl of YPD liquid in a 96-well plate and grown overnight at 30 °C shaking at 200 rpm until culture saturation. The distribution of the four replicates on the 96-well plate was optimised to reduce cross-contamination between wells and spot position effect involving bad distribution of compounds in the plate as well as border effects. A volume of 3 μL of saturated cell culture was diluted into 200 μL of sterile water. Then, 5 μL of diluted cells were spotted on plates containing YPD agar with additional fluconazole (2 μg/ml) or anidulafungin (0.125 μg/ml) in the case of the antifungal drug testing; concentrations of 0.5, 1 and 2 M NaCl for the salinity test and with either 5 mM DTT, 50 μg/ml CFW or 150 μg/ml SDS for the stress test. All transfers between plates were carried out using a 96-channel PlateMaster (Gilson). Spotted agar plates were then placed on scanners placed inside a 30 °C incubator, except for the temperature tests where the scanners were set either 24, 26, 30, 34, 35, 36 38 or 41.5 °C. Scanned images of the plates were taken every 15 min for a period of 24 h. Image processing as well as the calculation of growth rates and Area Under the Curve (AUC) were carried out with an in-house pipeline (https://github.com/Gabaldonlab/imageAnalysisPipeline_solid96wellPlates) based on the software Colonyzer[96]. Fitness Area Under the Curve (fAUC) values were calculated by dividing the AUC in the presence of either fluconazole or anidulafungin by the AUC in the absence of the drug (fAUC = $AUC_{antifungal}/AUC_{YPD}$). In the case of liquid medium, the 5 μL of diluted cells are added to 200 μL of YPD, which for the antifungal drug testing, contained additional fluconazole (2 μg/ml) or anidulafungin (0.125 μg/ml). Plates containing cells and drugs were then incubated for 24 h at 30 °C. For temperature tests, plates were incubated at either 30, 35, 38, 40 or 41.5 °C. Absorbance $OD_{600}$ values were read using a BioTek Synergy HTX microplate reader (Agilent). Plates were shaken for 5 s before reading.

### BSA degradation assay
The detection of secreted aspartyl proteases by *C. orthopsilosis* cells was performed using a BSA degradation assay largely as described in[40]. Briefly, cells were grown and prepared spotted on plates following the same procedure as for the phenotypic experiments (above section). Cells were plated on yeast carbon based medium (dextrose 20 g/L, $MgSO_4$ 0.5 g/L, $KH_2PO_4$ 1 g/L) supplemented with 0.2% BSA (bovine serum albumin). Plates were then incubated at 35 °C and appearance of a proteolytic halo was evaluated after 6 days.

### Virulence assays
*C. orthopsilosis* parental strains Co90-125 and SY36 as well as hybrid strains 2Y71 (environmental from clade 1) and MCO457 (clinical from clade 4) were grown overnight in 5 ml of YPD at 35ªC shaking at 200 rpm. From the saturated overnight cultures 2 ml were taken and washed twice with Phosphate Buffered Solution (PBS). *Galleria mellonella* larvae were reared in our laboratory following standardised breeding guidelines described in[97]. Larvae of similar size and weight were injected with 10 μl of a $10^8$ cells/ml dilution resulting in $10^6$ cells/larva. We injected 15 larvae per strain in two independent experiments. A second experiment with a lower inoculum of 5x10E5 cells/larva was also performed (Fig. S10). Survival of the larvae was monitored for 7 days. Assessment of survival was done by applying mechanical pressure to the larvae and evaluating their ability to move the head and tail. Dead larvae as well as webbing were removed daily. We used the Kaplan-Meyer estimate to compare the survival curves over time.

### Reporting summary
Further information on research design is available in the Nature Portfolio Reporting Summary linked to this article.

## Data availability
The whole genome sequencing data generated in this study has been deposited in the NCBI sequence read archive (SRA) database under accession code PRJNA767198. All *C. orthopsilosis* raw sequencing Illumina reads are under BioSample accessions SAMN21907760 to SAMN21907768. *C. orthopsilosis* ONT data is under BioSample accession SAMN21909361. Genome assembly of *C. orthopsilosis* parent B strain SY36 ASM2665022v1 is under BioSample accession SAMN21909361 [https://www.ncbi.nlm.nih.gov/datasets/genome/GCA_026650225.1/]. Additional sequencing libraries used in this study were retrieved from BioProjects PRJEB4430], PRJNA322245 and PRJNA520893[11,12,23]. Ortholog sequences are available in the PhylomeDB v5 database[94] [http://phylomedb.org]. Source data are provided with this paper.

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

## Acknowledgements

We thank the grant (NPRP 6-647-1-127) from the Qatar National Research Fund (a member of Qatar Foundation) to RF, for kindly sharing strains and data collected during that study. TG acknowledges support from the Spanish Ministry of Science and Innovation for grants PID2021-126067NB-I00, CPP2021-008552, PCI2022-135066-2, PRE2019-088193, PGC2018-099921-B-I00, and PDC2022-133266-I00, cofounded by ERDF "A way of making Europe"; from the Catalan Research Agency (AGAUR) SGR01551; from the European Union's Horizon 2020 research and innovation programme (ERC-2016-724173); from the Gordon and Betty Moore Foundation (Grant GBMF9742); from the "La Caixa" foundation (Grant LCF/PR/HR21/00737), from the Instituto de Salud Carlos III (IMPACT Grant IMP/00019 and CIBERINFEC CB21/13/00061- ISCIII-SGEFI/ERDF). Finally, this project also received funding from the European Union's Horizon 2020 research and innovation programme under the Marie Skłodowska-Curie grant agreement No 754433, to VdO.

## Author contributions

T.G., V.M. and VdO conceived the project. V.d.O., E.K., E.S. and J.C.N.R. performed the experiments. V.d.O. performed the computational analysis. R.F., A.A.M., TB – Provided typed strains as well as information regarding the sampling location for strains collected in the NPRP 6-647-1-127 study, critically revised the manuscript. T.G. supervised the project. V.d.O. and T.G. wrote the manuscript with input from all authors. All authors have read and accepted the manuscript.

## Competing interests

The authors declare no competing interest.
