## [Peer Review File · Nature Communications]

Origin of fungal hybrids with pathogenic potential from warm seawater environments.Reviewer #1 (Remarks to the Author):

This manuscript by del Olmo and colleagues reports on the analysis of the genomes of the first sequenced environmental *C. orthopsilosis* strains, confirming the hybrid nature of this species and - most importantly - identifying the missing parental B lineage. The authors also performed phenotypic analyses of both hybrid and parental strains and identified differences between them. This enhances our understanding of the pre-existing environmental adaptations that allowed *C. orthopsilosis* to colonise and infect the human host. Consequently, this study is a nice example providing a complete overview of the genomic evolution of one of the *Candida* species of medical importance, and opens up perspectives for investigating the evolutionary trajectory of other related pathogenic yeasts, such as *C. albicans* and *C. tropicalis* known to more frequently cause disease in humans. This manuscript is well-written, although editing of some sections and figure panel locations should improve the reading flow. The conclusions are generally supported by the presented data, but some confirmatory experiments are important to include - particularly those pertaining to the phenotypic analyses of the isolates. The discussion section is particularly interesting to read, and addresses some of the limitations of this study.

Specific comments:

1-I believe restructuring the results section and some panels of figure 1 should improve the reading flow. I propose moving supplementary figure 1 to the main figure 1 as a complement to panel 1A. The data in initial S1 figure are important. Figure 1 should only include panels 1A (with Fig S1 merged to panel 1A) and 1B. The results section entitled "Mitochondrial inheritance" should be merged with the results section entitled "Identification of *Candida orthopsilosis* parent B lineage" (Lines 149-175) as it provides additional arguments for assigning parent B lineage to SY36 and SY78.

2-I suggest to generate Figure 2 linked to section entitled "A reference genome assembly for *C. orthopsilosis* parent B and comparison with parent A" that describes the reference genome assembly features for *C. orthopsilosis* parent B and the comparative analyses with parent A. New Figure 2 should include initial panels 1C (becomes Fig2A), 1D (becomes Fig 2B) and the GO term enrichment analysis (as a graph with histograms of enriched categories) in panel 2C (fig 2C). The GO term enrichment results are important and reflect on potential functional features that could have contributed to the adaptation of the hybrid progeny.

3-With regard to parental B lineage assignment, preference was given to SY36. What is the explanation for the slightly lower homozygosity of strain SY78? This should be further detailed in lines 166-168.

4-Figure 5, described only in 2 lines (434-435) should be more detailed in the discussion section with a separate paragraph summarizing its impact.

5-Phenotypic assays were performed on solid agar medium only (YPD, supplemented or not with drugs/chemicals or incubated under different growth temperatures). These provide phenotypes reflecting a specific condition (solid medium). Authors should perform phenotyping in liquid medium as well, as growth in liquid versus solid media could lead to phenotypic differences. Testing both growth conditions (i.e. solid and liquid) is also relevant with regard to the environmental versus human body niches in which *C. orthopsilosis* may thrive.

Minor points, typos:

- Legend to figures 1-5 lacks details. Every single shape in the figure needs to be annotated. Same for supplementary figures.
- Add reference 12 (Schröder et al. 2016) to statements in lines 150-151 and 84-85.
- Lines 62-63, references 11 and 13 need to be switched here.
- Line 67: remove "s" in "hybrids".
- Lines 72-73: "THE vast majority... IS HYBRID".
- Line 262: comma between "their genome" and "hybrid".
- Line 264: correct "homEologous".

- Lines 270-271: Add reference to "confirming previous observations of four hybridization events".
- Line 682: "rpm" not "rmp"
- Line 686: 5x10E5 not printed correctly.

Reviewer #2 (Remarks to the Author):

In this manuscript, del Olmo et al. present a very nice and well conducted study that allows to shed light on the origin and the evolutionary history of a human-associated opportunistic pathogen, *Candida orthopsilosis*. The prevalence of hybrids in clinical strains was previously assessed, and one parental strains (parent A) already identified. Here, the authors sequenced and analyzed the first set of environmental strains, that were isolated from warm sea water. Despite the limited number of samples (9), their results suggest prevalence of hybrid strains in the marine environment too and 2 isolates were identified as from the 2nd parental lineage (parent B). Interestingly, the environmental isolates cluster in distinct clades, together with clinical isolates, which suggest several independent human colonizations events. They completed their study with phenotypic characterization, that show that the thermotolerance of all hybrids is most probably inherited from parent B, while the resistance to some drugs are specific to clinical isolates, suggesting rapid acquisition of resistance by exposure to the compounds. Their results are in line with the hypothesis that environment serves as a reservoir of opportunistic fungal pathogen species and that climate changes could be a driver in the emergence of thermotolerant pathogens.

The proposed analyses are elegant and convincing, and they support most of the conclusions. Few aspects would however need some clarifications:

- for the identification of parent B, the authors test the correspondence of the homozygous SNPs in the hybrids that were mapped vs parent A
the divergence between parent A and "potential B" could have been discussed more into detail before to better understand some of the given numbers.
Moreover, regarding the ~ 10% of positions that do not correspond to parent B, are these positions shared within/between the groups? Are they localized randomly along the genomes or enriched in subtelomeric or repeated regions? These data could inform about how diverse the SY36 strain and the actual parent B are and would be interesting to be discussed
- through comparison of the assemblies of parent A and YS36, some genomic regions were found to be differentially present. Are the genomic regions specifically detected in YS36 found in the hybrids? This would reinforce this isolate as being parent B or a very close relative.
- regarding the mitochondrial inheritance part: how can isolates from clade 1 and 3 harbor different mitotypes (namely mt1 and mt3) if they result from hybridization events between parent A and B? Could mt1 and mt3 also correspond to recombinant mitotypes? If not, what could be the hypothesis behind this phenomenon? This should be mentioned and discussed.
- the studied strains are most probably all diploid, but this is never mentioned in the manuscript. Moreover, have aneuploidies been explored at the whole species level and if so, could they be link to a specific clade/trait/...?
- "In order to explain this phenotype we assessed the genotype of heat-shock- and temperature-related proteins in the parental and hybrid strains but found no clear genetic difference that could explain the lack of thermotolerance of parental A strains" : has the copy number of these genes also been checked? Are there some LOH in favor of parent B in these regions? Are the intermediate phenotypes related to the allelic version the strains carry?

Minor comments:

- main figures 3 and 4 could be assembled into panels of the same figure
- fig S4 C/D: why are the strains not ordered similarly in both plots? Moreover, the number of heterozygous SNPs seems to be notably different for some strains while mapping vs parent A and B, which seems weird
- fig S6: for sake of clarity, clades should be indicated on this figure
- fig S7A: *C. glabrata* instead of CBS138
- page 8, 13.172.296 bp should be 13,172,296 bp. Numbers should be checked along the manuscript
- "the most different in terms of homo- and heterogeneity" should rather be "homo- and heterozygosity"
- "have too been isolated from marine environments" should rather be "have been isolated from marine environments too"

Reviewer #1 (Remarks to the Author):

This manuscript by del Olmo and colleagues reports on the analysis of the genomes of the first sequenced environmental *C. orthopsilosis* strains, confirming the hybrid nature of this species and - most importantly - identifying the missing parental B lineage. The authors also performed phenotypic analyses of both hybrid and parental strains and identified differences between them. This enhances our understanding of the pre-existing environmental adaptations that allowed *C. orthopsilosis* to colonise and infect the human host. Consequently, this study is a nice example providing a complete overview of the genomic evolution of one of the *Candida* species of medical importance, and opens up perspectives for investigating the evolutionary trajectory of other related pathogenic yeasts, such as *C. albicans* and *C. tropicalis* known to more frequently cause disease in humans. This manuscript is well-written, although editing of some sections and figure panel locations should improve the reading flow. The conclusions are generally supported by the presented data, but some confirmatory experiments are important to include - particularly those pertaining to the phenotypic analyses of the isolates. The discussion section is particularly interesting to read, and addresses some of the limitations of this study.

Response: We thank the reviewer for the nice and constructive comments on our work.

Specific comments:

1-I believe restructuring the results section and some panels of figure 1 should improve the reading flow. I propose moving supplementary figure 1 to the main figure 1 as a complement to panel 1A. The data in initial S1 figure are important. Figure 1 should only include panels 1A (with Fig S1 merged to panel 1A) and 1B. The results section entitled "Mitochondrial inheritance" should be merged with the results section entitled "Identification of *Candida orthopsilosis* parent B lineage" (Lines 149-175) as it provides additional arguments for assigning parent B lineage to SY36 and SY78.

Response: As suggested by Reviewer #1, supplementary figure 1 has been moved to main figure 1 and has been incorporated to panel 1A. Currently, figure 1 contains all k-mer plots of the marine isolates as well as the k-mer plots for parent A (Co90-125), a strain belonging to parental A lineage (s428) and a hybrid strain (MCO456) for comparison purposes. Similarly, as suggested by the reviewer, we have merged the section of Mitochondrial inheritance.

2-I suggest to generate Figure 2 linked to section entitled "A reference genome assembly for *C. orthopsilosis* parent B and comparison with parent A" that describes the reference genome assembly features for *C. orthopsilosis* parent B and the comparative analyses with parent A. New Figure 2 should include initial panels 1C (becomes Fig2A), 1D (becomes Fig 2B) and the GO term enrichment analysis (as a graph with histograms of enriched categories) in panel 2C (fig 2C). The GO term enrichment results are important and reflect on potential functional features that could have contributed to the adaptation of the hybrid progeny.

Response: As suggested by Reviewer #1, figure 2 is now linked to the section entitled "A reference genome assembly for *C. orthopsilosis* parent B and comparison with parent A".

Currently figure 2 contains a plot comparing the assemblies of parent A and B (panel 2A), a species tree with members of the *C. parapsilosis* species complex that shows the distinct placement of parent A, B and hybrid lineages of *C. orthopsilosis* (panel 2B) and a graph showing the results of GO term enrichment analysis of genes harboured in parent B specific regions. GO term enrichment in biological function is shown in main figure 2C. Graphs showing GO term enrichment in cell compartment and molecular function are shown in Supplemental 3 figure. The graphs were generated using ClusterProfiler R package which is now mentioned in lines 631-633.

3-With regard to parental B lineage assignment, preference was given to SY36. What is the explanation for the slightly lower homozygosity of strain SY78? This should be further detailed in lines 166-168.

Response: When mapped to the genome assembly of parent A, the strain SY36 showed a slightly higher number of homozygous SNPs as well as a lower number of heterozygous SNPs than strain SY76, thus the former was chosen as a representative for the parental B lineage. This is now further detailed in the manuscript text in lines 169-175. The difference in the amount of heterozygous variants between the two strains belonging to parent B lineage was a marginal 1,257 SNPs (present in SY76 and not in SY36). We studied the distribution of heterozygous variants between the two strains and found no significant differences, the variants were distributed evenly across the genome. Therefore we attribute the slightly higher heterozygosity of SY78 to the effects of mapping to a reference that is ~4% divergent and of variability due to variant calling algorithms.

4-Figure 5, described only in 2 lines (434-435) should be more detailed in the discussion section with a separate paragraph summarizing its impact.

Response: As suggested by Reviewer #1, Figure 5 has been commented more extensively in the discussion (lines 449-454).

5-Phenotypic assays were performed on solid agar medium only (YPD, supplemented or not with drugs/chemicals or incubated under different growth temperatures). These provide phenotypes reflecting a specific condition (solid medium). Authors should perform phenotyping in liquid medium as well, as growth in liquid versus solid media could lead to phenotypic differences. Testing both growth conditions (i.e. solid and liquid) is also relevant with regard to the environmental versus human body niches in which *C. orthopsilosis* may thrive.

Response: As suggested by Reviewer #1, we now also performed the phenotypic assays involving drug susceptibility and temperature tolerance (which were the most relevant ones in this study) in liquid medium. The strains chosen to perform this experiment were the same as for solid medium with antifungal drugs: Co90-125 (parent A), SY36 (parent B), 2Y71 (environmental hybrid from clade 1), MCO456 (clinical hybrid from clade 1), 2Y220 (environmental hybrid from clade 2), s435 (clinical hybrid from clade 2), 4Y222 (environmental hybrid from clade 3), 85 (clinical hybrid from clade 3), MCO457 and s498 (both clinical hybrids from clade 4 as there are no environmental isolates in this clade). The drug concentrations used were identical as for the experiment in solid medium: 0.125µg/ml of anidulafungin and 2µg/ml of fluconazole. For the temperature assay, we slightly reduced the number of tested

temperatures and performed the analysis only at the most relevant temperatures (from testing 24, 26, 30, 34, 35, 36 38 and 41.5 °C in solid to testing 30, 35, 38, 40 and 41.5 °C in liquid).

The experiment was performed in the exact same manner as with solid medium but instead of spotting 5µl of diluted cells onto the agar plate, they were added to 200µl of YPD liquid medium (with or without antifungal drugs) in a 96-well plate. Cells were then incubated at the corresponding temperature without shaking for 24 hours. After this period, the absorbance was measured at a wavelength of 600 nm (OD600). This experiment is now detailed in Materials and methods (lines 718-723) and the results can be visualised in Supplemental figure 7B and 8

The resulting trends of growth in liquid medium are, for the most part, identical to what we observed in solid media. When anidulafungin was present in the medium, and in agreement with what had been observed in solid medium, parent B strain SY36 was the least susceptible of all. Parent A was unable to grow and most hybrid strains showed an intermediate phenotype. In solid medium we observed that hybrids belonging to clades 2, 3 and 4 were less susceptible to anidulafungin than hybrids in clade 1, although the clade 1 hybrids were able to show a certain degree of growth. In liquid medium we observed the same trend with the difference that hybrid strains from clade 1, as well as strain 85 from clade 3, were unable to proliferate.

In the case of fluconazole, identical to what had been observed in solid medium, none of the environmental strains was able to grow. Parent A was able to grow, despite not being the least susceptible strain like in solid medium. Among hybrids, in agreement with the results in solid medium, clinical strains from clades 3 and 4 showed the best growth.

Regarding the temperature assays, all strains behaved in the exact same manner in liquid and in solid medium. Parent B was clearly the most thermotolerant in contrast to parent A which showed a strong impairment in growth at temperatures above (and including) 38°C. All hybrid strains showed an intermediate phenotype. We consider that these additional results provide further support to our phenotypic inference.

Minor points, typos:

- Legend to figures 1-5 lacks details. Every single shape in the figure needs to be annotated. Same for supplementary figures.
- Add reference 12 (Schröder et al. 2016) to statements in lines 150-151 and 84-85.
- Lines 62-63, references 11 and 13 need to be switched here.
- Line 67: remove "s" in "hybrids".
- Lines 72-73: "THE vast majority... IS HYBRID".
- Line 262: comma between "their genome" and "hybrid".
- Line 264: correct "homEologous".
- Lines 270-271: Add reference to "confirming previous observations of four hybridization events".
- Line 682: "rpm" not "rmp"
- Line 686: 5x10E5 not printed correctly.

Response: Thank you for noting these errors. All minor points have been addressed and are now implemented in the latest version of the manuscript.

Reviewer #2 (Remarks to the Author):

In this manuscript, del Olmo et al. present a very nice and well conducted study that allows to shed light on the origin and the evolutionary history of a human-associated opportunistic pathogen, *Candida orthopsilosis*. The prevalence of hybrids in clinical strains was previously assessed, and one parental strains (parent A) already identified. Here, the authors sequenced and analyzed the first set of environmental strains, that were isolated from warm sea water. Despite the limited number of samples (9), their results suggest prevalence of hybrid strains in the marine environment too and 2 isolates were identified as from the 2nd parental lineage (parent B). Interestingly, the environmental isolates cluster in distinct clades, together with clinical isolates, which suggest several independent human colonizations events. They completed their study with phenotypic characterization, that show that the thermotolerance of all hybrids is most probably inherited from parent B, while the resistance to some drugs are specific to clinical isolates, suggesting rapid acquisition of resistance by exposure to the compounds.

Their results are in line with the hypothesis that environment serves as a reservoir of opportunistic fungal pathogen species and that climate changes could be a driver in the emergence of thermotolerant pathogens.

The proposed analyses are elegant and convincing, and they support most of the conclusions. Few aspects would however need some clarifications:

Response: We thank the reviewer for the appreciation of our work and the helpful comments. We have made an effort to provide all requested clarifications in the revised manuscript.

- for the identification of parent B, the authors test the correspondence of the homozygous SNPs in the hybrids that were mapped vs parent A
the divergence between parent A and “potential B” could have been discussed more into detail before to better understand some of the given numbers.
Moreover, regarding the ~ 10% of positions that do not correspond to parent B, are these positions shared within/between the groups? Are they localized randomly along the genomes or enriched in subtelomeric or repeated regions? These data could inform about how diverse the SY36 strain and the actual parent B are and would be interesting to be discussed

Response: regarding the identification of parent B and the divergence with A, we now mention the previously measured allelic divergence in *C. orthopsilosis* hybrids in the introduction (line 86), and clarify the purpose in this comparison in more detail (lines 174-175).

Regarding the ~ 10% of positions that do not correspond to parent B, these are SNPs that are present in the hybrids but are absent in parent B. In other words, positions where there is variability in hybrids but where parent B does not diverge from parent A. We studied the distribution of these variants along the genomes of representative hybrids (two hybrids from

each clade). The distribution pattern was mostly clade-dependent although some variants were shared amongst clades. No particular accumulation or enrichment at subtelomeric regions was found. This is now mentioned in the reviewed version of the manuscript (lines 183-184).

- through comparison of the assemblies of parent A and YS36, some genomic regions were found to be differentially present. Are the genomic regions specifically detected in YS36 found in the hybrids? This would reinforce this isolate as being parent B or a very close relative.

Response: Genomic regions specific to potential parental B strain SY36 were found by mapping the Illumina sequencing reads of parent A Co90-125 to the assembly of SY36 and scanning for regions where Co90-125 showed no coverage. Given that these regions are distributed evenly across the genome, we assessed the genotype of six representative hybrid strains (two from each clade, one clinical and one environmental when possible) in the B-specific loci across chromosome 1. We observed that the proportion of regions in hybrids that harboured a B allele accounted for 53, 70, 75 and 81% of all B-specific regions for clades 1, 2, 3 and 4, respectively. The fact that genomic regions that are specifically detected in SY36 are also detected in hybrid strains, as suggested by Reviewer #2, reinforces the notion that strain SY36 is indeed parent B. These new results are mentioned in the reviewed manuscript (lines 224-230)

- regarding the mitochondrial inheritance part: how can isolates from clade 1 and 3 harbor different mitotypes (namely mt1 and mt3) if they result from hybridization events between parent A and B? Could mt1 and mt3 also correspond to recombinant mitotypes? If not, what could be the hypothesis behind this phenomenon? This should be mentioned and discussed.

Response: As reported in previous work by Schroder et al. 2016, mitotypes 1 and 3 (mt1 and mt3) were inherited from the parental A lineage. The only two recombinant mitotypes are those from strains belonging to nuclear clade 4 and harbour parts of mt4 (parent A) and mt2 (parent B). Strains from nuclear clade 2 inherited the mitotype (mt2) from parent B. This is currently mentioned in the latest version of the manuscript (lines 187-192). The explanation for this is that more than one mitotype coexists in parental A lineages and different hybridization events brought about different mitotypes into different clades, this is also mentioned in the text now.

- the studied strains are most probably all diploid, but this is never mentioned in the manuscript. Moreover, have aneuploidies been explored at the whole species level and if so, could they be link to a specific clade/trait/...?

Response: Ploidy levels have to be provided as input in the PerSVade pipeline used in this study for variant calling (-p 1/2/3). Therefore, despite not being mentioned in the initial manuscript, we previously analysed the ploidy of all *C. orthopsilosis* strains using a program based on base frequencies at variable sites (nQuire). The results showed that all hybrid strains were clearly diploid. Given their low levels of heterozygosity, the analysis was inconclusive for strains belonging to parental lineages although diploidy was assumed in those cases.

As suggested by Reviewer #2, this is currently explained in the latest version of the manuscript (lines 592-595). Moreover, we assessed aneuploidies with PerSVade but found no evidence of their presence in any of the isolates.

- “In order to explain this phenotype we assessed the genotype of heat-shock- and temperature-related proteins in the parental and hybrid strains but found no clear genetic difference that could explain the lack of thermotolerance of parental A strains” : has the copy number of these genes also been checked? Are there some LOH in favor of parent B in these regions? Are the intermediate phenotypes related to the allelic version the strains carry?

Response: Copy number variants and presence of LOH were analysed for all regions mentioned. No CNV that would explain the phenotypes were detected. There was also no enrichment of heterozygosity or LOH in the analysed loci. LOH in these regions was strongly clade dependent and, as observed for most of the genome, no preference towards parent A or B was shown for any of the hybrids. These observations are mentioned in the reviewed version of the manuscript (lines 315-319).

Minor comments:

- main figures 3 and 4 could be assembled into panels of the same figure
- fig S4 C/D: why are the strains not ordered similarly in both plots? Moreover, the number of heterozygous SNPs seems to be notably different for some strains while mapping vs parent A and B, which seems weird
- fig S6: for sake of clarity, clades should be indicated on this figure
- fig S7A: *C. glabrata* instead of CBS138
- page 8, 13.172.296 bp should be 13,172,296 bp. Numbers should be checked along the manuscript
- “the most different in terms of homo- and heterogeneity” should rather be “homo- and heterozygosity”
- “have too been isolated from marine environments” should rather be “have been isolated from marine environments too”

Response: Thank you for all these suggestions. All minor comments by Reviewer #2 have been addressed and are now implemented in the latest version of the manuscript.